# Is ChatGPT a Financial Expert? Evaluating Language Models on Financial Natural Language Processing

**Yue Guo    Zian Xu    Yi Yang**
The Hong Kong University of Science and Technology
yguoar@connect.ust.hk    zxubz@connect.ust.hk    imyiyang@ust.hk

## Abstract

The emergence of Large Language Models (LLMs), such as ChatGPT, has revolutionized general natural language preprocessing (NLP) tasks. However, their expertise in the financial domain lacks a comprehensive evaluation. To assess the ability of LLMs to solve financial NLP tasks, we present FinLMEval, a framework for Financial Language Model Evaluation, comprising nine datasets designed to evaluate the performance of language models. This study compares the performance of encoder-only language models and the decoder-only language models. Our findings reveal that while some decoder-only LLMs demonstrate notable performance across most financial tasks via zero-shot prompting, they generally lag behind the fine-tuned expert models, especially when dealing with proprietary datasets. We hope this study provides foundation evaluations for continuing efforts to build more advanced LLMs in the financial domain.

## 1 Introduction

Recent progress in natural language processing (NLP) demonstrates that large language models (LLMs), like ChatGPT, achieve impressive results on various general domain NLP tasks. Those LLMs are generally trained by first conducting self-supervised training on the unlabeled text (Radford et al., 2019; Brown et al., 2020; Touvron et al., 2023a) and then conducting instruction tuning (Wang et al., 2023; Taori et al., 2023) or reinforcement learning from human feedback (RLHF) (Ouyang et al., 2022) to let them perform tasks following human instructions.

Financial NLP, in contrast, demands specialized knowledge and specific reasoning skills to tackle tasks within the financial domain. However, for general language models like ChatGPT, their self-supervised training is performed on the text from various domains, and the reinforcement learning feedback they receive is generated by non-expert workers. Therefore, how much essential knowledge and skills are acquired during the learning process remains uncertain. As a result, a comprehensive investigation is necessary to assess its performance on financial NLP tasks.

To fill this research gap, we are motivated to evaluate language models on financial tasks comprehensively. For doing so, we propose a framework for Financial Language Model Evaluation (FinLMEval). We collected nine datasets on financial tasks, five from public datasets evaluated before. However, for those public datasets, it is possible that their test sets are leaked during the training process or provided by the model users as online feedback. To eliminate this issue, We used four proprietary datasets on different financial tasks: financial sentiment classification (FinSent), environmental, social, and corporate governance classification (ESG), forward-looking statements classification (FLS), and question-answering classification (QA) for evaluation.

In the evaluation benchmark, we evaluate the encoder-only language models with supervised fine-tuning, with representatives of BERT (Devlin et al., 2019), RoBERTa (Liu et al., 2019), FinBERT (Yang et al., 2020) and FLANG (Shah et al., 2022). We then compare the encoder-only models with the decoder-only models, with representatives of Chat-GPT (Ouyang et al., 2022), GPT-4 (OpenAI, 2023), PIXIU (Xie et al., 2023), LLAMA2-7B (Touvron et al., 2023b) and Bloomberg-GPT (Wu et al., 2023) by zero-shot prompting. Besides, we evaluate the efficacy of in-context learning of ChatGPT with different in-context sample selection strategies.

Experiment results show that (1) the fine-tuned task-specific encoder-only model generally performs better than decoder-only models on the financial tasks, even if decoder-only models have much larger model size and have gone through more pre-training and instruction tuning or RLHF; (2) when the supervised data is insufficient, the

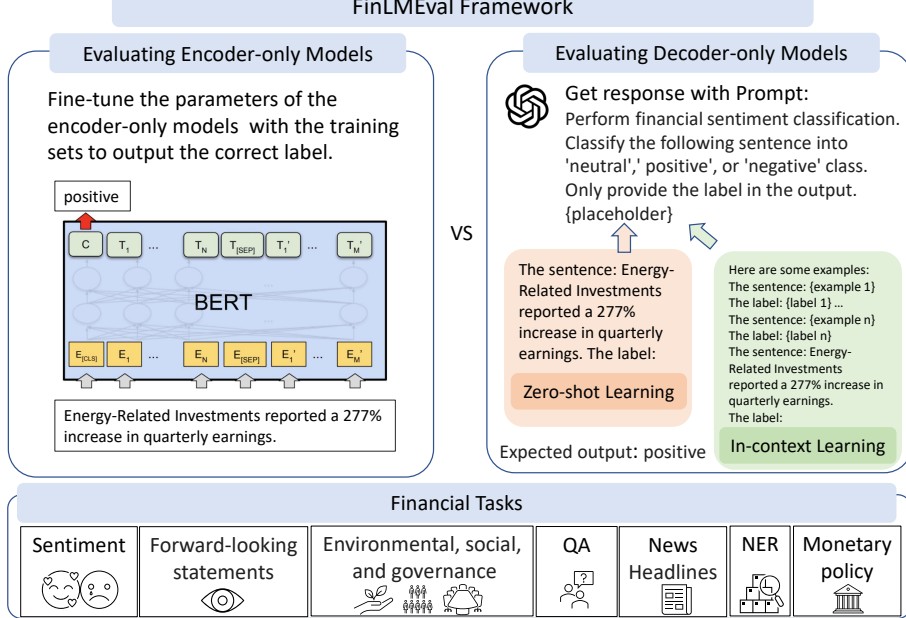

Figure 1: The framework of financial language model evaluation (FinLMEval).

zero-shot decoder-only models have more advantages than fine-tuned encoder-only models; (3) the performance gap between fine-tuned encoder-only models and zero-shot decoder-only models is more significant on private datasets than the publicly available datasets; (4) in-context learning is only effective under certain circumstances.

To summarize, we propose an evaluation framework for financial language models. Compared to previous benchmarks in the financial domain like FLUE (Shah et al., 2022), our evaluation includes four new datasets and involves more advanced LLMs like ChatGPT. We show that even the most advanced LLMs still fall behind the fine-tuned expert models. We hope this study contributes to the continuing efforts to build more advanced LLMs in the financial domain.

## 2 Related Works

The utilization of language models in financial NLP is a thriving research area. While some general domain language models, like BERT (Devlin et al., 2019), RoBERTa (Liu et al., 2019), GPT (Brown et al., 2020; OpenAI, 2023) and LLAMA (Touvron et al., 2023a,b) have been applied to financial NLP tasks, financial domain models like FinBERT (Araci, 2019; Yang et al., 2020; Huang et al., 2023), FLANG (Shah et al., 2022), PIXIU (Xie et al., 2023), InvestLM (Yang et al., 2023) and BloombergGPT (Wu et al., 2023) are specifically designed to contain domain expertise and generally

perform better in financial tasks. Recent work such as FLUE (Shah et al., 2022) has been introduced to benchmark those language models in the finance domain. However, the capability of more advanced LLMs, like ChatGPT and GPT-4, has not been benchmarked, especially on proprietary datasets. In this work, in addition to the public tasks used in FLUE, we newly include four proprietary tasks in FinLMEval and conduct comprehensive evaluations for those financial language models.

## 3 Methods

We compare two types of models in FinLMEval: the Transformers encoder-only models that require fine-tuning on the labeled dataset, and decoder-only models that are prompted with zero-shot or few-shot in-context instructions. Figure 1 provides an outline of evaluation methods of FinLMEval.

### 3.1 Encoder-only Models

Our experiments explore the performance of various notable encoder-only models: BERT (Devlin et al., 2019), RoBERTa (Liu et al., 2019), FinBERT (Yang et al., 2020) and FLANG (Shah et al., 2022). BERT and RoBERTa are pre-trained on general domain corpora, while FinBERT and FLANG are pre-trained on a substantial financial domain corpus. We fine-tune the language models on specific tasks. Following the fine-tuning process, inference can be performed on the fine-tuned models for specific applications.

| | # train | # test | source | description |
|---|---|---|---|---|
| FinSent | 8996 | 1000 | - | Financial sentiment classification dataset from analyst reports. |
| FPB | 2453 | 1000 | (Malo et al., 2014) | Sentiment classification dataset from financial news. |
| FiQA SA | 973 | 200 | (FiQA) | Aspect-based financial sentiment analysis. |
| ESG | 3000 | 1000 | - | Environmental, social, and corporate governance classification dataset. |
| FLS | 2600 | 1000 | - | Forward-looking statements classification dataset from corporate reports. |
| QA | 868 | 200 | - | Classification on the validity of question-answering pairs. |
| Headlines | 9570 | 1000 | (Sinha and Khandait, 2020) | Mulitple tasks classification dataset from news headlines. |
| NER | 14041 | 1000 | (Alvarado et al., 2015) | Named entity recognition on financial agreements. |
| FOMC | 1831 | 450 | (Shah et al., 2023) | Hawkish-dovish monetary policy classification from FOMC documents. |

Table 1: The summarization of nine datasets in FinLMEval. FPB, FiQA SA, Headlines, NER and FOMC are from public datasets, and FinSent, ESG, FLS and QA are newly collected and not released before.

## 3.2 Decoder-only Models

We also evaluate the performance of various popular decoder-only language models: ChatGPT (Ouyang et al., 2022), GPT-4 (OpenAI, 2023), PIXIU (Xie et al., 2023), LLAMA2-7B (Touvron et al., 2023b) and Bloomberg-GPT (Wu et al., 2023). ChatGPT and GPT-4, developed by OpenAI, are two advanced LLMs that showcase exceptional language understanding and generation abilities. The models are pre-trained on a wide array of textual data and reinforced by human feedback. PIXIU is a financial LLM based on fine-tuning LLAMA (Touvron et al., 2023a) with instruction data. LLAMA2 is a popular open-sourced LLM pre-trained on extensive online data, and BloombergGPT is an LLM for finance trained on a wide range of financial data. As the model size of the evaluated decoder-only models is extremely large, they usually do not require fine-tuning the whole model on downstream tasks. Instead, the decoder-only models provide answers via zero-shot and few-shot in-context prompting.

We conduct zero-shot prompting for all decoder-only models. We manually write the prompts for every task. An example of prompts for the sentiment classification task is provided in Figure 1, and the manual prompts for other tasks are provided in Appendix A. Furthermore, to evaluate whether few-shot in-context learning can improve the model performance, we also conduct in-context learning experiments on ChatGPT. We use two strategies to select the in-context examples for few-shot in-context learning: random and similar. The former strategy refers to random selection, and the latter selects the most similar sentence regarding the query sentence. All in-context examples are selected from the training set, and one example is provided from each label class.

## 4 Datasets

Our evaluation relies on nine datasets designed to evaluate the financial expertise of the models from diverse perspectives. Table 1 overviews the number of training and testing samples and the source information for each dataset. Below, we provide an introduction to each of the nine datasets.

**FinSent** is a newly collected sentiment classification dataset containing 10,000 manually annotated sentences from analyst reports of S&P 500 firms.

**FPB Sentiment Classification** (Malo et al., 2014) is a classic sentiment dataset of sentences from financial news. The dataset consists of 4840 sentences divided by the agreement rate of 5-8 annotators. We use the subset of 75% agreement.

**FiQA SA** (FiQA) is a aspect-based financial sentiment analysis dataset. Following the "Sentences for QA-M" method in (Sun et al., 2019), for each (sentence, target, aspect) pair, we transform the sentence into the form "what do you think of the {aspect} of {target}? {sentence}" for classification.

**ESG** evaluates an organization's considerations on environmental, social, and corporate governance. We collected 2,000 manually annotated sentences from firms' ESG reports and annual reports.

**FLS**, the forward-looking statements, are beliefs and opinions about a firm's future events or results. FLS dataset, aiming to classify whether a sentence contains forward-looking statements, contains 3,500 manually annotated sentences from the Management Discussion and Analysis section of annual reports of Russell 3000 firms.

**QA** contains question-answering pairs extracted from earnings conference call transcripts. The goal of the dataset is to identify whether the answer is valid to the question.

**Headlines** (Sinha and Khandait, 2020) is a dataset for the commodity market that analyzes

| Datasets | Encoder-only Models | | | | Decoder-only Models | | | | |
| --- | --- | --- | --- | --- | --- | --- | --- | --- | --- |
| | BERT | RoBERTa | FinBERT | FLANG-BERT | ChatGPT | GPT-4 | PIXIU | LLAMA2-7B | Bloomberg-GPT |
| FinSent | 0.841 | **0.871** | 0.851 | 0.849 | 0.782 | 0.809 | 0.800 | 0.243 | - |
| FPB | 0.914 | 0.934 | 0.912 | 0.881 | 0.869 | 0.905 | **0.965** | 0.339 | 0.511 |
| FiQA SA | 0.750 | 0.875 | 0.805 | 0.695 | 0.898 | 0.920 | **0.930** | 0.480 | 0.751 |
| ESG | 0.931 | 0.956 | **0.958** | 0.925 | 0.477 | 0.626 | 0.509 | 0.209 | - |
| FLS | 0.875 | 0.862 | **0.882** | 0.861 | 0.652 | 0.565 | 0.275 | 0.365 | - |
| QA | **0.865** | 0.825 | 0.825 | 0.785 | 0.695 | 0.775 | 0.680 | 0.625 | - |
| Headlines-PDU | 0.937 | 0.947 | **0.956** | 0.940 | 0.889 | 0.878 | 0.842 | 0.411 | - |
| Headlines-PDC | 0.978 | 0.979 | **0.981** | 0.978 | 0.936 | 0.947 | 0.702 | 0.053 | - |
| Headlines-PDD | 0.954 | **0.961** | 0.960 | 0.956 | 0.896 | 0.900 | 0.763 | 0.382 | - |
| Headlines-PI | 0.974 | 0.964 | 0.976 | **0.977** | 0.225 | 0.105 | 0.753 | 0.966 | - |
| Headlines-AC | 0.996 | 0.993 | **0.997** | 0.995 | 0.806 | 0.838 | 0.902 | 0.346 | - |
| Headlines-FI | 0.976 | 0.964 | 0.976 | 0.974 | 0.711 | 0.780 | **0.981** | 0.048 | - |
| Headlines-PS | 0.905 | 0.918 | **0.924** | 0.906 | 0.630 | 0.811 | 0.776 | 0.546 | - |
| NER | 0.980 | **0.981** | 0.964 | 0.978 | 0.748 | 0.707 | 0.749 | 0.714 | 0.608 |
| FOMC | 0.587 | 0.611 | 0.602 | 0.602 | 0.633 | **0.729** | 0.522 | 0.349 | - |
| Average | 0.897 | **0.909** | 0.905 | 0.907 | 0.723 | 0.753 | 0.739 | 0.405 | - |

Table 2: The results of fine-tuned encoder-only models and zero-shot decoder-only models in 9 financial datasets. The results, except the NER dataset, are measured in micro-F1 score. NER is measured in accuracy. Although some zero-shot decoder-only models can achieve considerate results in most cases, the fine-tuned encoder-only models usually perform better than decoder-only models.

news headlines across multiple dimensions. The tasks include the classifications of Price Direction Up (PDU), Price Direction Constant (PDC), Price Direction Down (PDD), Asset Comparison(AC), Past Information (PI), Future Information (FI), and Price Sentiment (PS).

**NER** (Alvarado et al., 2015) is a named entity recognition dataset of financial agreements.

**FOMC** (Shah et al., 2023) aims to classify the stance for the FOMC documents into the tightening or the easing of the monetary policy.

Among the datasets, FinSent, ESG, FLS, and QA are newly collected proprietary datasets.

# 5 Experiments

This section introduces the experiment setups and reports the evaluation results.

## 5.1 Model Setups

**Encoder-only models setups.** We use the BERT (base,uncased), RoBERTa (base), FinBERT (pre-train), and FLANG-BERT from Huggingface[1], and the model fine-tuning is implemented via Trainer [2]. For all tasks, we fix the learning rate as $2 \times 10^{-5}$, weight decay as 0.01, and the batch size as 48. We randomly select 10% examples from the training set as the validation set for model selection and

fine-tune the model for three epochs. Other hyper-parameters remain the default in Trainer.

**Decoder-only models setups.** In the zero-shot setting, for ChatGPT and GPT-4, We use the "gpt-3.5-turbo" and "gpt-4" model API from OpenAI, respectively. We set the temperature and top_p as 1, and other hyperparameters default by Ope-nAI API. The ChatGPT results are retrieved from the May 2023 version, and the GPT-4 results are retrieved in August 2023. For PIXIU and LLAMA2, we use the "ChanceFocus/finma-7b-nlp" and "meta-llama/Llama-2-7b" models from Huggingface. The model responses are generated greedily. All prompts we used in the zero-shot setting are shown in Appendix A. Besides, as the BloombergGPT (Wu et al., 2023) is not publicly available, we directly adopt the results from the original paper.

For in-context learning, we conduct two strategies for in-context sample selection: random and similar. We select one example from each label with equal probability weighting for random sample selection. For similar sample selection, we get the sentence embeddings by SentenceTransformer (Reimers and Gurevych, 2019) "all-MiniLM-L6-v2" model[3] and use cosine similarity as the measure of similarity. Then, we select the sentences with the highest similarity with the query sentence as the

---

[1] https://huggingface.co/

[2] https://huggingface.co/docs/transformers/main_classes/trainer

[3] https://www.sbert.net/

| Datasets | ChatGPT | | |
|---|---|---|---|
| | zero | ic-ran | ic-sim |
| FinSent | 0.782 | 0.761 | 0.761 |
| FPB | 0.869 | 0.832 | 0.844 |
| FiQA SA | 0.898 | 0.891 | 0.891 |
| ESG | 0.477 | 0.726 | 0.800 |
| FLS | 0.652 | 0.673 | 0.636 |
| QA | 0.695 | 0.660 | 0.675 |
| Headlines-PDU | 0.889 | 0.839 | 0.765 |
| Headlines-PDC | 0.936 | 0.323 | 0.413 |
| Headlines-PDD | 0.896 | 0.816 | 0.788 |
| Headlines-PI | 0.225 | 0.768 | 0.844 |
| Headlines-AC | 0.806 | 0.576 | 0.597 |
| Headlines-FI | 0.711 | 0.606 | 0.592 |
| Headlines-PS | 0.630 | 0.690 | 0.729 |
| NER | 0.748 | 0.784 | 0.793 |
| FOMC | 0.633 | 0.672 | 0.650 |
| Average | **0.723** | 0.708 | 0.719 |

Table 3: The results of ChatGPT in zero-shot and in-context few-shot learning. Zero, ic-ran, and ic-sim represent zero-shot learning, in-context learning with random sample selection, and in-context learning with similar sample selection. The zero-shot and few-shot performances are comparable in most cases.

in-context examples. The prompts for in-context learning are directly extended from the corresponding zero-shot prompts, with the template shown in Figure 1.

### 5.2 Main Results

Table 2 compares the results of the fine-tuned encoder-only models and zero-shot decoder-only models in 9 financial datasets. We have the following findings:

**In 6 out of 9 datasets, fine-tuned encoder-only models can perform better than decoder-only models.** The decoder-only models, especially those that have experienced RLHF or instruction-tuning, demonstrate considerable performance on zero-shot settings on the financial NLP tasks. However, their performance generally falls behind the fine-tuned language models, implying that these large language models still have the potential to improve their financial expertise. On the other hand, **fine-tuned models are less effective when the training examples are insufficient** (FiQA SA) **or imbalanced** (FOMC)**.**

**The performance gaps between fine-tuned models and zero-shot LLMs are larger on pro-**prietary datasets than publicly available ones.** For example, the FinSent, FPB, and FiQA SA datasets are comparable and all about financial sentiment classification. However, zero-shot LLMs perform the worst on the proprietary dataset FinSent. The performance gaps between fine-tuned models and zero-shot LLMs are also more significant on other proprietary datasets (ESG, FLS, and QA) than the public dataset.

Table 3 compares the zero-shot and in-context few-shot learning of ChatGPT. In ChatGPT, **the zero-shot and few-shot performances are comparable in most cases**. When zero-shot prompting is ineffective, adding demonstrations can improve ChatGPT's performance by clarifying the task, as the results of ESG and Headlines-PI tasks show. Demonstrations are ineffective for easy and well-defined tasks, such as sentiment classifications and Headlines (PDU, PDC, PDD, AC, and FI), as the zero-shot prompts clearly instruct ChatGPT.

## 6 Conclusions

We present FinLMEval, an evaluation framework for financial language models. FinLMEval comprises nine datasets from the financial domain, and we conduct the evaluations on various popular language models. Our results show that fine-tuning expert encoder-only models generally perform better than the decoder-only LLMs on the financial NLP tasks, and adding in-context demonstrations barely improves the results. Our findings suggest that there remains room for enhancement for more advanced LLMs in the financial NLP field. Our study provides foundation evaluations for continued progress in developing more sophisticated LLMs within the financial sector.

## 7 Limitations

This paper has several limitations to improve in future research. First, our evaluation is limited to some notable language models, while other advanced LLMs may exhibit different performances from our reported models. Also, as the LLMs keep evolving and improving over time, the future versions of the evaluated models can have different performance from the reported results. Second, FinLMEval only focuses on financial classification tasks, and the analysis of the generation ability of the LLMs still needs to be included. Future work can be done toward developing evaluation benchmarks on generation tasks in the financial domain.

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
