# OpenReview forum: "Is ChatGPT a Financial Expert?  Evaluating Language Models on Financial Natural Language Processing"
_EMNLP/2023/Conference — EMNLP 2023 Findings_

### Official Review · Reviewer_VqpE · 2023-08-03

**Soundness:** 2

**Excitement:**

3: Ambivalent: It has merits (e.g., it reports state-of-the-art results, the idea is nice), but there are key weaknesses (e.g., it describes incremental work), and it can significantly benefit from another round of revision. However, I won't object to accepting it if my co-reviewers champion it.

**Paper Topic And Main Contributions:**

This paper presents FinLMEval, an evaluation framework for financial language models (LLMs). FinLMEval comprises 9 datasets in financial domain. The authors also conducted the evaluations on fine-tuning auto-encoding models and zero/few-shot prompting ChatGPT. Current results show that the fine-tuning auto-encoding models generally perform better than ChatGPT.

**Questions For The Authors:**

Questions:
1.	There are tough financial NLP tasks, such as financial technical report generation, season financial report generation, market analysis by financial news analysis. These are true financial NLP tasks, and simple sentiment analysis or NER are far from enough for real-world financial NLP tasks using LLMs. Wish to see more evaluation on these directions. Also, if the authors have any feedbacks and ideas in these directions, willing to learn as well.

**Reasons To Accept:**

Strong:
1.	FinLMEval helps evaluating financial LLMs in a relatively fair way;

**Reasons To Reject:**

Weak:
1.	This paper is more fit as a demonstration paper, and the major contribution can hardly be concluded (existing datasets’ collection, evaluation, and comparison are more experimental).

**Reproducibility:**

3: Could reproduce the results with some difficulty. The settings of parameters are underspecified or subjectively determined; the training/evaluation data are not widely available.

**Reviewer Confidence:**

4: Quite sure. I tried to check the important points carefully. It's unlikely, though conceivable, that I missed something that should affect my ratings.

---

> ### Author Rebuttal · Authors · 2023-08-29
>
> Dear Reviewer VqpE:
>
> Thank you for your review of our work. We truly appreciate the time you took to evaluate our work. Your insights and feedback are invaluable to us.
>
> We would like to give some explanations on the raised points:
>
> 1. Response to reason to reject:
>
> Although the method in our paper is experimental, our work provides an empirical study of how well the language models perform on financial tasks. We address the research gap that, though the GPT-like general domain LLMs show unprecedented ability on general NLP tasks, their ability on financial tasks remains unclear. Our work aims to conduct an empirical study to fill this gap.
>
> Our contributions are summarized as (1) We draw interesting conclusions from the empirical study. For example, we find that the fine-tuned expert models are usually better than the general generative LMs, especially on the private datasets; when the labeled data is insufficient or imbalanced, general generative LMs can perform better. Our empirical study can guide future researchers and domain practitioners to choose suitable LMs when facing related problems. (1) We provide a new tool and new financial evaluation benchmark. Four datasets (FinSent, FLS, ESG, QA) are newly included in our evaluation tools.
>
> 2. Response to question for the authors:
>
> We agree that most of our tasks are based on classifications, which can only evaluate the LMs from one perspective; additional generative tasks on the financial domain are needed.
> Nevertheless, compared to the previous FLUE benchmarks (Shah et al., 2022), we include four new document classification datasets, which require domain expert knowledge to understand the financial concepts and judge the financial reports, which is an improvement from the previous datasets.
>
> Future works can focus on evaluating financial LMs on generative tasks. Potential directions could be: (1) construct datasets on financial generation tasks, e.g., financial report generation/summarization, financial news analysis, financial knowledge question-answering, etc., and (2) evaluation tools to align LMs’ responses with financial experts’ preferences.
> If you have any further questions or suggestions for collaboration, please feel free to reach out. We are always open to fruitful discussions and potential partnerships.
>
> Once again, thank you for your support and feedback. We look forward to sharing more of our work with you in the future.
>
> Best regards,
>
> Authors

---

### Official Review · Reviewer_nbqJ · 2023-08-06

**Soundness:** 4

**Excitement:**

3: Ambivalent: It has merits (e.g., it reports state-of-the-art results, the idea is nice), but there are key weaknesses (e.g., it describes incremental work), and it can significantly benefit from another round of revision. However, I won't object to accepting it if my co-reviewers champion it.

**Paper Topic And Main Contributions:**

This paper proposes a new evaluation framework for financial natural language processing, which mainly aims to evaluate how much domain knowledge is learned by the recently emerged LLMs (mostly ChatGPT). Designed prompts corresponding to different tasks and new originally annotated dataset are provided.

**Questions For The Authors:**

Question A: would the collected dataset be public? It would be helpful for the community to reproduce and develop new models.

**Reasons To Accept:**

- Filling the gap that the financial understanding ability of ChatGPT is under-explored.

- Comprehensive experiment setting for the GPT-like generative models:

  - Design prompts to implement zero-shot and in-context learning protocols on ChatGPT to evaluate.

  - Collect extra datasets on 4 tasks to verify the LLM ability on private corpora.

**Reasons To Reject:**

- Misleading Taxonomy: "auto-encoding" models, i.e. encoder-only models, should never be treated as a parallel concept of "LLM". The authors misuse the terminology encoder-only, decoder-only, and encoder-decoder architectures here. Such categorisation can be found in Wang, Hongyu, et al. Foundation Transformers. arXiv:2210.06423, arXiv, 19 Oct. 2022. arXiv.org, https://doi.org/10.48550/arXiv.2210.06423.
- Missing Baselines:
  - Following the taxonomy mentioned above, the authors should provide the evaluation results from other generative pre-trained LM baselines (eg. T5, BART) to see whether difference in the results are brought by the distinguished architecture. If authors also want to use models with similar parameters and training data sizes to ChatGPT , they can alternatively provide results from GLM or FLAN-T5.
  - Since this work is an incremental version of FLUE benchmark (shah et al 2022), the baseline models (FLANG-BERT, FLANG-ELECTREA) should be included for comparison.

**Reproducibility:**

4: Could mostly reproduce the results, but there may be some variation because of sample variance or minor variations in their interpretation of the protocol or method.

**Reviewer Confidence:**

4: Quite sure. I tried to check the important points carefully. It's unlikely, though conceivable, that I missed something that should affect my ratings.

**Typos Grammar Style And Presentation Improvements:**

line#39: => are performed on text from various domains

Two “consequently” are used consecutively in line#42 and line#44. Do the author meant to emphasise a chain-of-thought?

line#47 "because of this": "this" is vague in the context.

---

> ### Author Rebuttal · Authors · 2023-08-29
>
> Dear Reviewer nbqJ:
>
> Thank you for your review of our work. We truly appreciate the time you took to evaluate our work. Your insights and feedback are invaluable to us.
>
> We would like to give some explanations on the raised points:
>
> 1. Response to reason 1 to reject:
>
> We apologize for the misleading taxonomy. Our paper is intended to evaluate the BERT-like encoder-only models as fine-tuned expert models and GPT-like decoder-only models as general language models. We will modify the paper by using the terms encoder- and decoder-only models more professionally.
>
> 2. Response to reason 2 to reject:
>
> Thanks for raising the suggestion that some baselines still need to be included. To have a more comprehensive evaluation of other advanced LMs, we include the experimental results on decoder-only models ChatGPT (August 3 Version), GPT-4, PIXIU (a newly released financial LLM), LLAMA, LLAMA2, and the encoder-only financial language models FLANG-BERT and FLANG-ELECTRA. The results are shown below.
>
> |               | Previous best results | FLANG-BERT | FLANG-ELECTRA | ChatGPT, August | GPT-4 |      PIXIU      | LLAMA-7B | LLAMA2-7B |
> |---------------|:---------------------:|:----------:|:-------------:|:---------------:|:-----:|:---------------:|:--------:|:---------:|
> | FinSent       |    **0.871(RoBERTa)**    |    0.849   |     0.853     |      0.751      | 0.809 | 0.800   |   0.528  |   0.243   |
> | FPB           |     0.934(RoBERTa)    |    0.881   |     0.784     |      0.755      | 0.905 |      **0.965**      |   0.119  |   0.339   |
> | FiQA SA       |  0.898(ChatGPT, May)  |    0.695   |     0.695     |      0.769      | 0.920 |      **0.930**      |   0.310  |   0.480   |
> | ESG           |     **0.958(FinBERT)**    |    0.925   |     0.914     |      0.643      | 0.626 |      0.509      |   0.190  |   0.209   |
> | FLS           |     **0.882(FinBERT)**    |    0.861   |     0.864     |      0.566      | 0.565 |      0.275      |   0.343  |   0.365   |
> | QA            |      **0.865(BERT)**     |    0.785   |     0.820     |      0.705      | 0.775 |      0.680      |   0.720  |   0.625   |
> | Headlines-PDU |     **0.956(FinBERT)**    |    0.940   |     0.948     |      0.848      | 0.878 |      0.842      |   0.405  |   0.411   |
> | Headlines-PDC |     **0.981(FinBERT)**   |    0.978   |     0.980     |      0.786      | 0.947 |      0.702      |   0.053  |   0.053   |
> | Headlines-PDD |    **0.961(RoBERTa)**   |    0.956   |     0.954     |      0.866      | 0.900 |      0.763      |   0.384  |   0.382   |
> | Headlines-PI  |     0.976(FinBERT)    |   **0.977**   |     0.963     |      0.195      | 0.105 |      0.753      |   0.966  |   0.966   |
> | Headlines-AC  |     **0.997(FinBERT)**   |    0.995   |     0.815     |      0.763      | 0.838 |      0.902      |   0.318  |   0.346   |
> | Headlines-FI  |  0.976(BERT,FinBERT)  |    0.974   |     0.977     |      0.795      | 0.780 |     **0.981**    |   0.083  |   0.048   |
> | Headlines-PS  |    **0.924(FinBERT)**    |    0.906   |     0.918     |      0.773      | 0.811 |      0.776      |   0.460  |   0.546   |
> | NER           |     **0.981(RoBERTa)**    |    0.978   |     0.978     |      0.654      | 0.707 |      0.749      |   0.815  |   0.714   |
> | FOMC          |   0.633(ChatGPT,May)  |    0.602   |     0.480     |      0.596      | **0.729** |      0.522      |   0.251  |   0.349   |
> | Average       |    **0.909(RoBERTa)**   |    0.907   |     0.890     |      0.698      | 0.753 |      0.739      |   0.396  |   0.405   |
>
> After comparing more language models, the main conclusions in the paper still hold, i.e.,
> (1)   The fine-tuned expert models are usually better than the general generative LMs. Their performance is improved after fine-tuning the generative LMs on the financial datasets (PIXIU model).
> (2)   The performance gaps between fine-tuned expert models and the general generative models are more significant on private than publicly available datasets.
>
> 3. Response to the question for the authors
>
> The release of the data is restricted by copyright concerns of the financial reports. Upon acceptance, we will build a financial leaderboard to evaluate the open LMs without explicitly releasing the data.
>
> If you have any further questions or suggestions for collaboration, please feel free to reach out. We are always open to fruitful discussions and potential partnerships.
>
> Once again, thank you for your support and feedback. We look forward to sharing more of our work with you in the future.
>
> Best regards,
>
> Authors

---

### Official Review · Reviewer_Fsxu · 2023-08-13

**Soundness:** 4

**Excitement:**

3: Ambivalent: It has merits (e.g., it reports state-of-the-art results, the idea is nice), but there are key weaknesses (e.g., it describes incremental work), and it can significantly benefit from another round of revision. However, I won't object to accepting it if my co-reviewers champion it.

**Paper Topic And Main Contributions:**

This paper explore the proficiency of ChatGPT in the financial domain, and introduce a tool called "FinLMEval" for evaluating ChatGPT on financial tasks.

The main contributions of this paper as following:
1. The author proposes FinLMEval, a comprehensive evaluation framework for assessing the performance of language models on financial natural language processing (NLP) tasks. FinLMEval comprises nine datasets specifically designed to evaluate the expertise of language models in the financial domain.
2. The authors compare the performance of fine-tuned auto-encoding language models (BERT, RoBERTa, FinBERT) and the large language model (LLM) ChatGPT on financial NLP tasks, and reveal that while ChatGPT performs well across most financial tasks, it generally lags behind the fine-tuned expert models, especially on proprietary datasets.
3. The authors identifies factors that affect the performance of language models on financial NLP tasks. It shows that fine-tuned task-specific models generally outperform ChatGPT, except when the supervised data is insufficient or imbalanced.

**Questions For The Authors:**

Q: While there were access restrictions in place for GPT-4 previously, there remains a question as to why the authors did not assess the performance of other Language Models. Solely conducting tests on ChatGPT could potentially compromise the accuracy and authenticity of the experimental findings.

**Reasons To Accept:**

This paper explore the proficiency of ChatGPT in the financial domain, and introduce a tool called "FinLMEval" for evaluating ChatGPT on financial tasks, and get some interesting conclusion as following:

1. While ChatGPT performs well across most financial tasks, it generally lags behind the fine-tuned expert models, especially on proprietary datasets.
2. The fine-tuned task-specific models generally outperform ChatGPT, except when the supervised data is insufficient or imbalanced.

Additional, the author proposes FinLMEval, a comprehensive evaluation framework for assessing the performance of language models on financial natural language processing (NLP) tasks. FinLMEval comprises nine datasets specifically designed to evaluate the expertise of language models in the financial domain.

**Reasons To Reject:**

1. The authors focuses solely on comparing ChatGPT with fine-tuned auto-encoding models and does not include a comparison with other advanced LLMs, such as GPT-4, LLAMA and Alpaca. So I think the contributions are not enough.
2. The related work section briefly mentions previous language models used in financial NLP tasks but does not provide a comprehensive overview of the existing literature.
3. The authors briefly mentions the evaluation strategies used for ChatGPT (zero-shot and few-shot in-context learning) but does not provide sufficient details about the experimental setup and methodology.

**Reproducibility:**

5: Could easily reproduce the results.

**Reviewer Confidence:**

4: Quite sure. I tried to check the important points carefully. It's unlikely, though conceivable, that I missed something that should affect my ratings.

---

> ### Author Rebuttal · Authors · 2023-08-29
>
> Dear Reviewer Fsxu:
>
> Thank you for your review of our work. We truly appreciate the time you took to evaluate our work. Your insights and feedback are invaluable to us.
>
> We would like to give some explanations on the raised points:
>
> 1. Response to reason 1 to reject:
>
> Thanks for raising the suggestion that our comparison is not enough. To have a more comprehensive evaluation of other advanced generative LLMs, we include the experimental results on decoder-only models ChatGPT (August 3 Version), GPT-4, PIXIU (a newly released financial LLM), LLAMA, LLAMA2, and the encoder-only financial language models FLANG-BERT and FLANG-ELECTRA. The results are shown as below.
>
> |               | Previous best results | FLANG-BERT | FLANG-ELECTRA | ChatGPT, August | GPT-4 |      PIXIU      | LLAMA-7B | LLAMA2-7B |
> |---------------|:---------------------:|:----------:|:-------------:|:---------------:|:-----:|:---------------:|:--------:|:---------:|
> | FinSent       |    **0.871(RoBERTa)**    |    0.849   |     0.853     |      0.751      | 0.809 | 0.800   |   0.528  |   0.243   |
> | FPB           |     0.934(RoBERTa)    |    0.881   |     0.784     |      0.755      | 0.905 |      **0.965**      |   0.119  |   0.339   |
> | FiQA SA       |  0.898(ChatGPT, May)  |    0.695   |     0.695     |      0.769      | 0.920 |      **0.930**      |   0.310  |   0.480   |
> | ESG           |     **0.958(FinBERT)**    |    0.925   |     0.914     |      0.643      | 0.626 |      0.509      |   0.190  |   0.209   |
> | FLS           |     **0.882(FinBERT)**    |    0.861   |     0.864     |      0.566      | 0.565 |      0.275      |   0.343  |   0.365   |
> | QA            |      **0.865(BERT)**     |    0.785   |     0.820     |      0.705      | 0.775 |      0.680      |   0.720  |   0.625   |
> | Headlines-PDU |     **0.956(FinBERT)**    |    0.940   |     0.948     |      0.848      | 0.878 |      0.842      |   0.405  |   0.411   |
> | Headlines-PDC |     **0.981(FinBERT)**   |    0.978   |     0.980     |      0.786      | 0.947 |      0.702      |   0.053  |   0.053   |
> | Headlines-PDD |    **0.961(RoBERTa)**   |    0.956   |     0.954     |      0.866      | 0.900 |      0.763      |   0.384  |   0.382   |
> | Headlines-PI  |     0.976(FinBERT)    |   **0.977**   |     0.963     |      0.195      | 0.105 |      0.753      |   0.966  |   0.966   |
> | Headlines-AC  |     **0.997(FinBERT)**   |    0.995   |     0.815     |      0.763      | 0.838 |      0.902      |   0.318  |   0.346   |
> | Headlines-FI  |  0.976(BERT,FinBERT)  |    0.974   |     0.977     |      0.795      | 0.780 |     **0.981**    |   0.083  |   0.048   |
> | Headlines-PS  |    **0.924(FinBERT)**    |    0.906   |     0.918     |      0.773      | 0.811 |      0.776      |   0.460  |   0.546   |
> | NER           |     **0.981(RoBERTa)**    |    0.978   |     0.978     |      0.654      | 0.707 |      0.749      |   0.815  |   0.714   |
> | FOMC          |   0.633(ChatGPT,May)  |    0.602   |     0.480     |      0.596      | **0.729** |      0.522      |   0.251  |   0.349   |
> | Average       |    **0.909(RoBERTa)**   |    0.907   |     0.890     |      0.698      | 0.753 |      0.739      |   0.396  |   0.405   |
>
> After comparing more language models, the main conclusions in the paper still hold, i.e.,
>
> (1)	The fine-tuned expert models are usually better than the general generative LMs. After fine-tuning the generative LMs on the financial datasets (PIXIU model), their performance is improved.
>
> (2)	The performance gaps between fine-tuned expert models and the general generative models are more significant on private datasets than the publicly available datasets.
>
> 2. Response to reason 2 to reject
>
> We apologize for not providing a comprehensive literature review due to the space limit of the short paper.  Some previous works developed language models for the finance domain. FinBERT (Araci, 2019) further pre-trains BERT on the financial domain corpus, while  FinBERT (Yang et al., 2020) collects a large-scale financial communication corpus to pre-train a financial domain-specific BERT model. Also, FinBERT (Liu et al.,2020) uses multi-task pre-training to simultaneously train the model on general corpora and financial domain corpora. Besides, FLANG (Shah et al., 2022) apply better masking using financial keywords and phrases with span boundary and in-filing objectives. Recently, several large generative financial language models have been developed. BloombergGPT (Wu et al., 2023) is a 50-billion parameter language model trained on a wide range of financial data. PIXIU (Xie et al., 2023) further fine-tunes LLaMA with financial instruction data. While many financial language models are proposed, their comprehensive evaluation needs to be addressed. FLUE (Shah et al., 2022) introduced to benchmark those language models in the finance domain. However, it mainly focuses on the encoder-only models, and the performance of the generative decoder-only model is not evaluated. Adding four extra proprietary financial datasets, we conduct a comprehensive empirical analysis and comparison of the performance of various financial language models.
>
> 3. Response to reason 3 to reject:
>
> We introduce the experimental setup in section 5.1. We prompt ChatGPT via the API provided by OpenAI, and the prompts for zero-shot learning on different datasets are provided in Appendix A. For in-context learning, " The sentence: {}" in prompts is replaced by " Here are some examples: The sentence: {sentence 1} The label: {label 1} … The sentence: {sentence n} The label: {label n} The sentence: {} The label:", as shown in figure 1.
>
> 4. Response to question 1 for the authors
>
> See response 1.
>
> If you have any further questions or suggestions for collaboration, please feel free to reach out. We are always open to fruitful discussions and potential partnerships.
>
> Once again, thank you for your support and feedback. We look forward to sharing more of our work with you in the future.
>
> Best regards,
>
> Authors
>
>
> References:
>
> Dogu Araci. 2019. Finbert: Financial sentiment analysis with pre-trained language models. CoRR,abs/1908.10063.
>
> Zhuang Liu, Degen Huang, Kaiyu Huang, Zhuang Li, and Jun Zhao. 2020. Finbert: A pre-trained financial language representation model for financial text mining. In IJCAI, pages 5–10.
>
> Raj Shah, Kunal Chawla, Dheeraj Eidnani, Agam Shah, Wendi Du, Sudheer Chava, Natraj Raman, Charese Smiley, Jiaao Chen, and Diyi Yang. 2022. When FLUE meets FLANG: Benchmarks and large pre-trained language model for financial domain. In EMNLP pages 2322–2335.
>
> Shijie Wu, Ozan Irsoy, Steven Lu, Vadim Dabravolski, Mark Dredze, Sebastian Gehrmann, Prabhanjan Kam-badur, David Rosenberg, and Gideon Mann. 2023. Bloomberggpt: A large language model for finance. arXiv preprint arXiv:2303.17564
>
> Qianqian Xie, Weiguang Han, Xiao Zhang, Yanzhao Lai, Min Peng, Alejandro Lopez-Lira, and Jimin Huang. 2023. PIXIU: A large language model, instruction data and evaluation benchmark for finance. CoRR, abs/2306.05443.
>
> Yi Yang, Mark Christopher Siy Uy, and Allen Huang.2020. Finbert: A pretrained language model for financial communications. CoRR, abs/2006.08097.

---

### Meta-Review · Area_Chair_JUKE · 2023-09-19

**Recommendation:** 3

**Metareview:**

The paper presents FinLMEval, an evaluation framework for assessing the performance of language models on financial natural language processing tasks. The authors conduct experiments to compare the performance of ChatGPT with fine-tuned auto-encoding models like BERT, RoBERTa, and FinBERT. The paper addresses a gap in the literature by focusing on the financial understanding abilities of large language models like ChatGPT. Reviewers mention that the paper could have been more comprehensive by including other large language models for comparison. There are also questions about the soundness of the methodology, especially regarding the specific NLP tasks chosen for evaluation. In light of the identified merits and shortcomings, the collective judgment leans toward accepting the paper for the "Findings" section. The manuscript makes a sound yet incremental contribution to the field, and its limitations could be addressed in future work.

---

### Decision · Program_Chairs · 2023-10-07

**Decision:**

Accept-Findings

**Comment:**

The paper presents FinLMEval, an evaluation framework for assessing the performance of language models on financial natural language processing tasks. The authors conduct experiments to compare the performance of ChatGPT with fine-tuned auto-encoding models like BERT, RoBERTa, and FinBERT. The paper addresses a gap in the literature by focusing on the financial understanding abilities of large language models like ChatGPT. Reviewers mention that the paper could have been more comprehensive by including other large language models for comparison. There are also questions about the soundness of the methodology, especially regarding the specific NLP tasks chosen for evaluation. In light of the identified merits and shortcomings, the collective judgment leans toward accepting the paper for the "Findings" section. The manuscript makes a sound yet incremental contribution to the field, and its limitations could be addressed in future work.